# Experimental Study of Abrasive Waterjet Cutting for Managing Residues in No-Tillage Techniques

**Francesco Perotti** [1,*] , **Massimiliano Annoni** [1] , **Aldo Calcante** [2] , **Michele Monno** [1] , **Valerio Mussi** [3] and **Roberto Oberti** [2]

1   Dipartimento di Meccanica, Politecnico di Milano, Via La Masa 1, 20156 Milan, Italy;
    massimiliano.annoni@polimi.it (M.A.); michele.monno@polimi.it (M.M.)
2   Department of Agricultural and Environmental Sciences, Università degli Studi di Milano, Via Celoria 2,
    20133 Milan, Italy; aldo.calcante@unimi.it (A.C.); roberto.oberti@unimi.it (R.O.)
3   Consorzio MUSP, Strada Torre della Razza, 29122 Piacenza, Italy; valerio.mussi@musp.net
*   Correspondence: francesco.perotti@polimi.it

**Abstract:** A laboratory investigation of abrasive waterjet cutting of wheat straws was conducted. The work was aimed at a systematic characterization of the abrasive waterjet cutting capability of wheat straws, as a potential alternative to cutting discs currently adopted in no-till drills and planters for crop residue management. A two level $2_{IV}^{7-3}$ fractional factorial design was applied to investigate the influence of abrasive waterjet process parameters on the cutting efficiency of wheat straws. Straw coverage thickness, water pressure, and orifice diameter were found to be the most significant ones. Experimental results suggest that straw cutting mechanism is mostly related to the hydraulic power of the jet. A multiple logistic regression was performed to model the relationship between the cutting efficiency and the jet power. The logistic model was then applied to estimate the average water and power consumption for wheat straw cutting during a no-tillage seeding operation. An average jet hydraulic power of 6400 W would be sufficiently high to guarantee 90% cutting efficiency in presence of heavy residue distribution. The experimental study shows that a small quantity of abrasive powder (50 g·min$^{-1}$) allows one to increase the jet cutting capability of wheat straws, and to reduce the required maximum hydraulic power, compared to pure waterjet cutting. Results show are potentially relevant for field validation in agriculture based on no-tillage.

**Keywords:** abrasive waterjet cutting; residue management; no-tillage technique





## 1. Introduction

Conservation agriculture is a farming system that promotes minimum soil disturbance, maintenance of a permanent soil cover, and diversification of plant species [1]. It is aimed at conserving and improving soil fertility, at reaching a high water-use efficiency in rainfed crops, and at reducing energy requirements for crop establishment. In CA, practice ground cover by crop residues plays a key role in limiting soil erosion by wind and water and in retaining soil moisture with reduced water evaporation [2]. Direct seeding, which is the most consistent technique with CA principles, is performed leaving all plant residues on the soil surface except along the seeding row where residues are cut and moved on the seeding furrow sides. For this, no-till equipment has to meet specific requirements. As described by [3] in direct seeders, each planter unit includes specific soil engaging tools that must be able to cut the soil through large quantities of residues, penetrate undisturbed soil, and deposit seed and fertilizer at a suitable depth 25–50 mm, depending on the crop to be sown. Tool cutting capability is affected by the intrinsic variability of soil properties (texture, moisture, and soil strength), surface roughness, and residue distribution. This, in general, results in an increase of required draft force and vertical force and consequently in higher tractor power and weight requirement compared to conventional seeders [4].

Direct seeding equipment specifically includes a residue management unit (RMU), i.e., a tine-type or more often a disc-type cutting tool [5]. The RMU function is to cut surface residues and push them sideways with respect to the seeding path, minimizing soil structure disturbance [4]. However, in presence of heavy surface residue, different problems occur during the seeding operation: residues tend to accumulate on the tools, causing clogging and malfunctioning, leading to a significant reduction of cutting efficiency; tool blockage; high cutting forces; and low reliability, especially in moist soil. Inadequate residue management ultimately results in a poor uniform seeding depth and seed deposition, leading to poor germination and inadequate crop establishment [6,7]. These disadvantages present opportunities for alternative methods. A new approach may be based on high-pressure waterjet technology that is one of the most widespread advanced manufacturing processes, used in industrial cutting of a wide range of materials [8–10] thanks to its flexibility of application and reduced cutting forces [11]. During the process, water is pressurized up to 300–400 MPa and passes through a primary orifice with a diameter usually in the range of 0.2–0.4 mm. The pressure energy of the water flow is converted into kinetic energy, resulting into high-speed waterjet as shown in Figure 1.

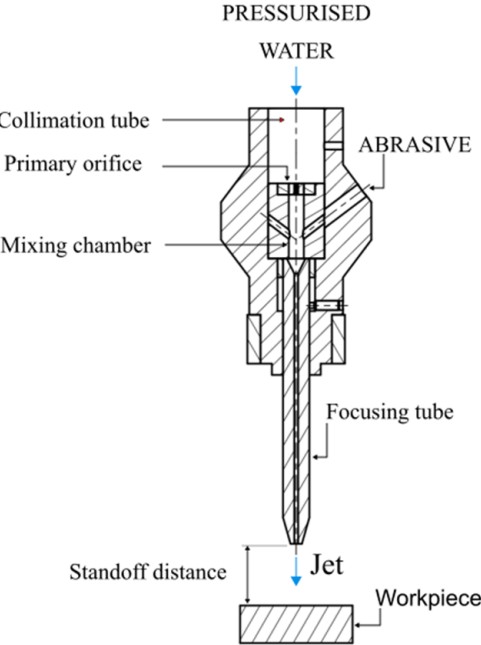

**Figure 1.** Abrasive waterjet cutting head, adapted from [12], with permission from publisher.

Waterjet velocity $v_j$ Equation (3) can be obtained from the theoretical velocity $v_{th}$ Equation (1) stated by the Bernoulli equation Equation (1) considering water compressibility $\psi$ Equation (2) and friction effects $c_v$, respectively [13].

$$v_{th} = \sqrt{\frac{2p}{\rho}} \tag{1}$$

$$\psi = \sqrt{\frac{L}{p(1-n)}\left[\left(1 + \frac{p}{L}\right)^{1-n} - 1\right]} \tag{2}$$

where $L = 300$ MPa and $n = 0.1368$ are empirical coefficients, measured for water [13].

$$v_j = c_v \psi \sqrt{\frac{2p}{\rho}} \tag{3}$$

Water mass flow rate $\dot{m}_w$ Equation (4) is determined from water density $\rho$ and water volume flow rate $Q_w$ Equation (5), which in turn can be calculated from the Bernoulli velocity $v_{th}$, the nominal cross-sectional area of the orifice $S_n$, and the orifice discharge coefficient $c_d$ [13]. The jet hydraulic power $P_{hydr}$ is defined in Equation (6).

$$\dot{m}_w = \rho Q_w \tag{4}$$

$$Q_w = c_d S_n \sqrt{\frac{2p}{\rho}} \tag{5}$$

$$P_{hydr} = \frac{1}{2}\dot{m}_w v_j^2 \tag{6}$$

Waterjet technology has been adopted in various manufacturing sectors as a cutting tool for a wide range of materials such as paper, wood, food, and soft materials. On the other hand, superior cutting capacity can be obtained with abrasive waterjet (AWJ) (Figure 1), i.e., by adding abrasive particles to the waterjet. AWJ is widely used as a manufacturing technology for ductile and brittle materials such as metals, composites, glass, and ceramics [11]. Waterjet cutting ability strongly depends on process parameters and on the physical properties of the processed material, which are reported in (Figure 2). In view of many advantages, AWJ technology has been used for cutting a wide variety of materials in different industrial sectors. However, its diffusion in the agricultural sector is quite limited and, even though AWJ technology has been cited as potential cutting technology for agricultural operations, the number of significant investigations in the literature is very small.

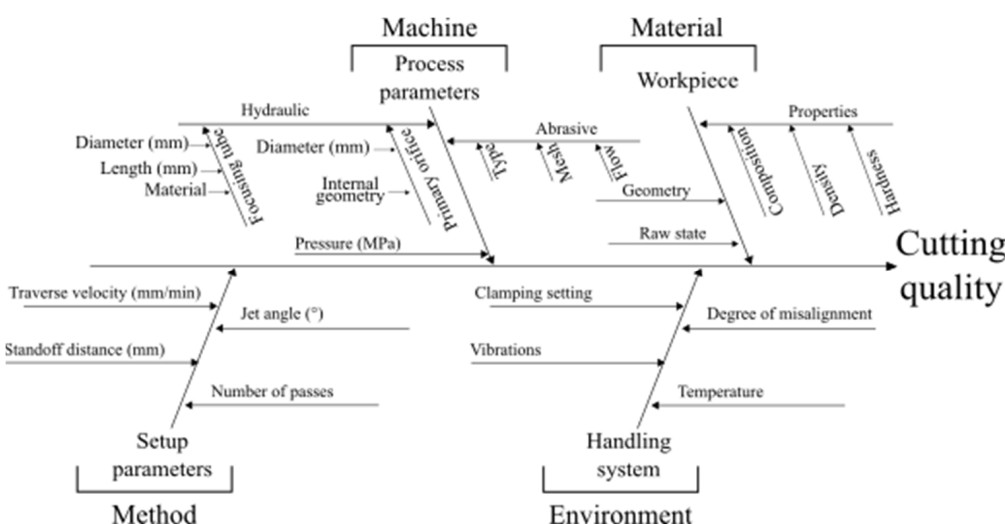

**Figure 2.** Ishikawa diagram of AWJ cutting process.

### 1.1. State of the Art

The literature review has shown that waterjet technology can be potentially involved in different agricultural tasks and processes, including soil opening for seeding and fertilizing processes, agricultural product cutting, and residue cutting for conservation agriculture. Pure waterjet cutting has been investigated as a cutting or peeling process to prevent agricultural products damage during minimal processing of raw materials [14,15]. Potentially, this method might be an alternative to blade and kinives in view of its many advantages, such as reduced rate of microbial contamination and oxidation, as well as its increased quality assurance [16–19]. Suspended ice particles [20–22], and salt particles [23], were mixed in waterjets, as abrasives, to increase the jet cutting capability of difficult-to-cut materials in the food sector.

Waterjet technology has been exploited as an innovative tool for liquid fertilizer injection, in contrast to broadcasting method, which causes a significant nitrogen loss through volatilization [24]. In [25], a liquid biofertiliser was pressurized up to 4.5 MPa, then it passed through a nozzle, whose diameter was 0.99 mm. The jet penetration capability of soil was tested in both laboratory and field studies. However, results showed that jet was not able to penetrate deeper than 0–20 mm, since the working pressure was too low. In [26], the authors proved the capability of PWJ in soil opening is a series of laboratory experiments. The effect of water pressure and water volume flow rate on injection depth in soil was investigated. Results showed that a water pressure of 40 MPa and a water flow rate of 7.5 L·min$^{-1}$ were sufficient to achieve an injection depth of 70–90 mm, at a traverse velocity of 2 m·s$^{-1}$.

Preliminary studies were performed to investigate the feasibility of waterjet cutting of crop residues ([27–29]) through a series of laboratory and field experiments, while numerical simulation of the process was investigated in [30,31]. Specifically, one of the main challenges was the identification of the most influential process parameters on jet cutting capability, to finally define the efficiency of waterjet technology. Table 1 shows literature focused on the quantity of cutting efficiency.

**Table 1.** Process parameters review.

| Author | $p$ (MPa) | $d_\mathrm{n}$ (mm) | $sod$ (mm) | $v$ (m·s$^{-1}$) | Residue | Response Variable |
|---|---|---|---|---|---|---|
| [27] | 200–400 | 0.23, 0.28, 0.36 | 30–230 | 0.44–1.33 | Sugarcane | Cutting depth |
| [28] | 170–380 | 0.15–0.30 | 5–70 | 1.67–3.33 | Wheat straws | Cutting capacity |
| [29] | 240–280 | 0.30 | 20 | 0.83–1.38 | Maize stalks | Cutting depth |

Valco [27] investigated pure waterjet cutting as an alternative method for cutting CP 44–101 sugarcane stalks, with a series of cutting experiments, in which sugarcanes specimens were arranged on a cutting plate and moved up to 1.33 m·s$^{-1}$, under a stationary waterjet. The waterjet process parameters were changed: water pressure, standoff distance (i.e., the distance between the orifice outlet and the specimen), traverse speed, primary orifice diameter, and cutting depth were measured after each test. The response variable was defined as the ratio of the measured depth by the average sugarcane stalk diameter (process parameters are summarized in Table 1). Results showed that the higher the pressure the better the cutting efficiency, but at a diminishing rate due to the decrease in jet coherence. The standoff distance to orifice diameter ratio ($sod/d_\mathrm{n}$) was found to be one of the most significant parameters influencing the response variable, as the cutting capability of the jet was found to be uniform up to $sod/d_\mathrm{n} \approx 400$, (80–100%) since the waterjet kinetic energy was sufficiently high to cut through the target material. Increasing the standoff distance, the waterjet started to break up, resulting in a lower kinetic energy as well as in a lower cutting capability. The higher the standoff distance, the lower the cutting efficiency. At large values of standoff distance, the jet coherence sensibly decreased until the jet broke up, resulting in a reduced cutting ability (20%). The hydrodynamic properties of the jet highly influence the cutting performance, which in turn depends on the water pressure and primary orifice diameter. A similar conceptual result was found by [28], in which the effect of both technological parameters and residue related parameters were investigated on the waterjet cutting efficiency of wheat straws. Specifically, the water pressure, primary orifice diameter, and traverse velocity were included in the former category (Table 1), while straw moisture level and arrangement were considered for the latter. The experimental layout was like that used in [27]. Wheat straws were arranged inside a holding that was moved up to 3.33 m·s$^{-1}$ under the waterjet. After each cut, straws were collected and divided in two categories, according to the degree of separation achieved, i.e., 0: no separation, 1: full separation. The jet cutting capability was calculated as weighted average of cut straws, considering full and partial separation, where each weight is the grade of separation (0, 0.5, 0.7, 0.9, and 1.0). A Hyplex pump (Flow International Corporation) was used to pressurize

water up to 380 MPa. Regardless of either the straw arrangement or moisture content, an increase in water pressure resulted in a higher cutting efficiency, and the same trend was observed by increasing the orifice diameter. However, the experimental design did not include a systematic investigation of waterjet process parameters effect on cutting efficiency; neither were interactions between potentially significant factors considered. Indeed, the standoff distance range of values appears to be too small for the purpose of process feasibility evaluation. For this reason, a deeper analysis would be required. In [29], an innovative ultrahigh-pressure waterjet-assisted furrow opener was designed and tested in a field experiment for the no-till management of maize stalks. An optimization analysis was proposed, to find process parameters that maximize the cutting capability. The optimal ranges were water pressure between 267 and 280 MPa, jet impingement angle between 80.2° and 90°, and forward speed between 1.11 and 2.22 m·s$^{-1}$. Within these ranges, stalks cut-off ratio could reach a level above 95% and no blockage of furrow opener occurred. The survey of the literature showed that no studies had focused on the abrasive waterjet cutting of crop residues.

### 1.2. Objectives

Given the lack of experimental studies on abrasive waterjet (AWJ) cutting of crop residue, the aim of this study is to investigate the effect of AWJ cutting process parameters on the cutting efficiency of wheat straws, in view of a potential application in no-till equipment. The conducted experiments aim to characterize the AWJ cutting process on a dense and thick layer of wheat straws, simulating the crop residues lying on the soil surface. To achieve this aim, a test bench system was developed allowing one to control the process parameters, as well as to test different experimental conditions. The first part of the work is related to the identification of the most influential process parameters and to the definition of the processability window, determined in a set of experiments planned conducted according to design of experiments (DOE) methodology. The second part of the study is focused on the investigation of the cutting mechanism, in connection to the hydraulic power of the jet. Finally, a preliminary evaluation regarding the impact of the technological solution in terms of feasibility development and consumptions is included.

## 2. Materials and Methods

### 2.1. Material and Sample Preparation

Wheat straws were collected, and internode trait of 209.0 ± 1.4 mm length was obtained for each straw; the average outer diameter was 4.8 ± 0.3 mm. A sample of sixty dry wheat straws is shown in Figure 3a, while Figure 3b shows the sample distribution of the outer diameter. Since the main objective of the study was to investigate the feasibility of AWJ cutting of heavy surface residue, samples were prepared consisting of wheat straw assemblies that modelled an ordered spatial distribution of straws in a field, as well as their stratification.

Straws were arranged inside a 3D printed PLA finned template (54 mm × 30 mm × 30 mm), to constrain both of the straw sides and to prevent straws disturbance during the cutting (Figure 3b). Straws were stacked one above the other, inside the free space between adjacent fins, until the target thickness was reached (e.g., 15 mm, 25 mm). This arrangement was equivalent to a residue surface density range of 6–10 Mg·ha$^{-1}$, which represents a heavy residue condition, in which many standard residue management units normally used in no-till approach fail to cut.

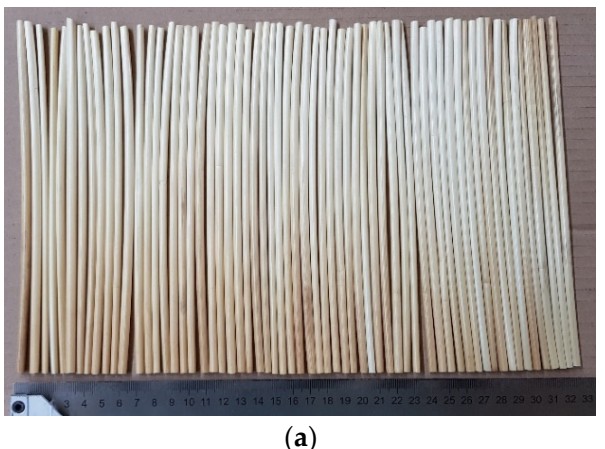

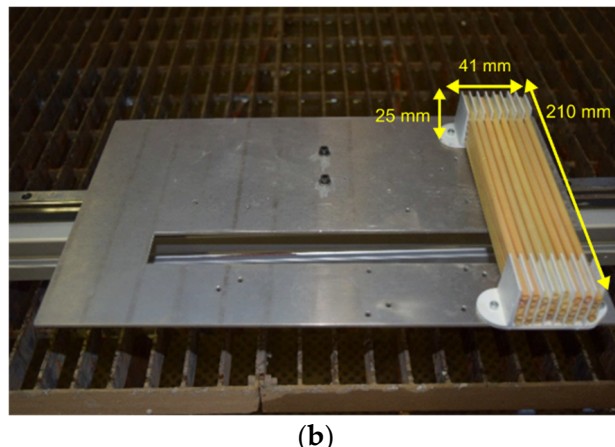

(**a**)  (**b**)

**Figure 3.** (**a**) Dry wheat straws. (**b**) Straws sample.

### 2.2. Experimental Equipment: AWJ Cutting System and Handling System

In this work, a CNC abrasive waterjet machine (IDRO 1740, Cms SpA, Zogno (Bg)) with a double effect high pressure intensifier pump (Flow International Corporation) was used. The operating maximum pressure was 300 MPa, with a power requirement of 30 kW, while the maximum water flow rate was 4.0 L·min$^{-1}$ A pressure gauge was mounted at the pressure intensifier outlet to measure water pressure in every experimental run. Abrasive mass flow rate was regulated by an abrasive dosing system. The maximum abrasive mass flow was 400 g·min$^{-1}$. A custom linear drive system was designed and built to move the straws samples to a speed up to 2.78 m·s$^{-1}$, for the purpose of simulating actual field velocities, during the direct seeding process. The developed system was a CNC linear actuator, based on belt transmission (Figure 4), with a travel distance of 1200 mm. An aluminium cutting table (60 mm × 150 mm in size) was mounted on the handling system, and it was designed to hold wheat straws samples. A central cavity (30 mm × 100 mm × 3 mm) was realized to avoid any interaction between the cutting plate and AWJ during the cutting experiments.

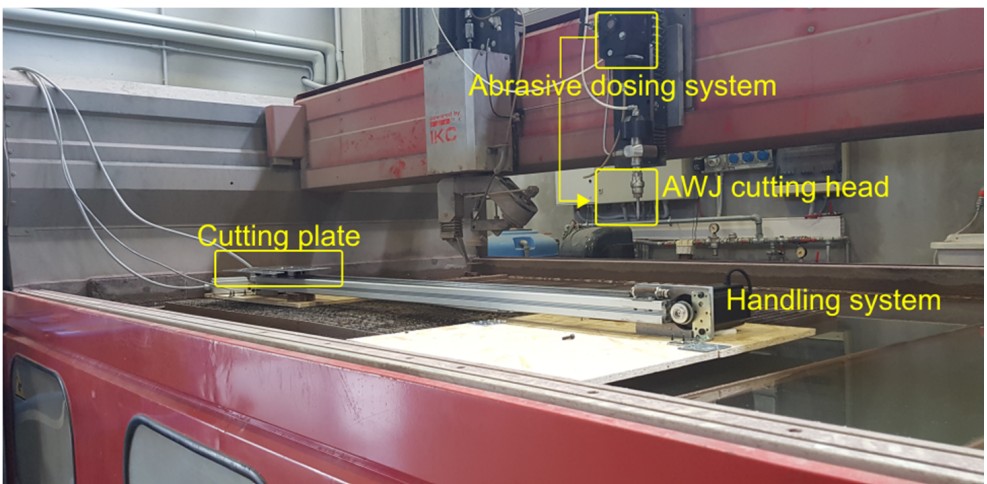

**Figure 4.** Experimental layout.

In this study, GMA GARNET (Australian GMA Garnet), mesh 80, was used as abrasive powder, which is composed by silicates and it is extracted from mines. Chemical and mineral composition is reported in Table 2.

**Table 2.** Chemical composition of the Garnet abrasive.

| Chemical Name | Symbol | Proportion (Weight %) |
|---|---|---|
| Almandine Garnet | $Fe_3Al_2(SiO_4)_3$ | >97 |
| Ilmenite | $FeTiO_3$ | <2.0 |
| Calcium Carbonate | $CaCo_3$ | <1.5 |
| Zircon | $ZrSiO_4$ | <0.2 |
| Quartz | $SiO_2$ | <0.2 |

*2.3. Experiments*

The experiments consisted in a series of cutting tests that were performed in one single pass on assembled straws samples. For each experimental run, the sample was moved under a stationary AWJ. The experiments were organized in two distinct steps: a first preliminary investigation and an experimental plan, based on DOE methodology. Due to the limited knowledge about AWJ cutting of wheat straws, a pre-experimental champaign was performed, with the aim of determining a processability window of the AWJ cutting. The following process parameters were considered:

- Primary orifice diameter, $d_n$ (mm);
- Water pressure, $p$ (MPa);
- Standoff distance, *sod* (mm);
- Abrasive mass flow rate, $\dot{m}_a$ (g·min$^{-1}$);
- Traverse velocity, $v$ (m·s$^{-1}$);
- Orientation, $\vartheta$ (°);
- Thickness, $t$ (mm).

In this study, the sample thickness (Figure 3a) represents the straw coverage, whilst the sample orientation is defined by the angle formed by straw axial direction and traverse velocity direction. The standoff distance is the distance from the focusing tube outlet to the sample. Preliminary cutting tests were performed on samples having a thickness of 15 mm, 210 mm length, and 40 mm wide. Assuming uniform coverage of straw distribution and a straw mass per unit length of 1.0 kg·m$^{-1}$, the above straw coverage condition would correspond to a realistic field density of about 6.0 Mg·ha$^{-1}$. To find the lower bound of process parameters range, different combinations of the process parameters were tested until a separation cut of one layer was achieved. This condition was assumed as a minimum requirement for the cutting process to achieve a significant result. A range of 50–200 g ·min$^{-1}$ of abrasive mass flow rate was selected as representative for usual waterjet technological applications [8]. The levels established for both fixed and variable factors are listed in Table 3, while the sequence of preliminary cutting tests is reported in Table 4.

**Table 3.** Constant parameters and variable factors of the experimental design.

| Factors | Value |
|---|---|
| Constant Parameters | |
| Impact angle, $\varphi$ (°) | 90 |
| Focusing tube length, $l_f$ (mm) | 75 |
| Type of abrasive | Barton Garnet |
| Abrasive mesh | 80 |
| Water pressure, $p$ (MPa) | 100–260 |
| Standoff distance, *sod* (mm) | 50–100 |
| Abrasive mass flow rate $\dot{m}_a$ (g·min$^{-1}$) | 25–200 |
| Traverse velocity, $v$ (m·s$^{-1}$) | 1.1–2.78 |
| Thickness, $t$ (mm) | 15–25 |
| Orientation, $\vartheta$ (°) | 60–90 |
| Primary orifice diameter, $d_n$ (mm) | 0.25–0.33 |
| Focusing tube diameter, $d_f$ (mm) | 0.75–1.0 |

**Table 4.** Preliminary experimental plan.

| Run Order | $d_n$ | $p$ | $sod$ | $\dot{m}_a$ | $v$ | $t$ | $\vartheta$ |
|---|---|---|---|---|---|---|---|
| 1 | 0.25 | 100 | 100 | 25 | 2.78 | 15 | 90 |
| 2 | 0.25 | 150 | 100 | 25 | 2.78 | 15 | 90 |
| 3 | 0.25 | 150 | 100 | 25 | 2.78 | 15 | 90 |
| 4 | 0.25 | 150 | 100 | 25 | 2.20 | 15 | 90 |
| 5 | 0.25 | 175 | 100 | 100 | 2.20 | 15 | 90 |
| 6 | 0.25 | 175 | 100 | 100 | 2.20 | 15 | 90 |
| 7 | 0.25 | 200 | 100 | 25 | 2.20 | 15 | 90 |
| 8 | 0.25 | 200 | 100 | 100 | 2.20 | 15 | 90 |
| 9 | 0.25 | 200 | 100 | 100 | 2.78 | 15 | 90 |
| 10 | 0.25 | 200 | 100 | 50 | 2.20 | 15 | 90 |
| 11 | 0.25 | 200 | 100 | 100 | 2.20 | 15 | 90 |

Starting from a tentative pressure value of 100 MPa, the traverse velocity was varied between 2.78 m·s$^{-1}$ and 2.2 m·s$^{-1}$; the abrasive mass flow rate was varied between 25 g·min$^{-1}$ and 100 g·min$^{-1}$; and standoff distance, sample orientation, and primary orifice diameter were held constant, according to the preliminary experimental plan (Table 4).

The second step of the experiment was based on a statistical design of experimental design, which aimed to investigate the process parameters' significance in terms of cutting efficiency $\varepsilon$. The last was defined in Equation (7), where $n_{cs}$ is the number of completely cut straws and $n_{us}$ the number of partially cut straws.

$$\varepsilon = \frac{n_{cs}}{n_{cs} + n_{us}} \tag{7}$$

Since there are seven parameters, the experiment was performed according to a single-replicate, two-level $2_{IV}^{7-3}$ fractional factorial design. The primary orifice diameter was set as a blocking factor, since past experiments [32] showed that changing the cutting head configuration (e.g., primary orifice) between experimental runs resulted in a nuisance factor [33]. The sixteen corner points of the experimental design were divided into two blocks, corresponding to 0.25 mm and 0.33 mm of primary orifice diameter. The experimental runs were randomized within each block. The focusing tube was selected according to the optimal ratio, between the primary orifice diameter and the focusing tube diameter such that $d_f/d_n \cong 3$ [32]. A 1.0 mm and 0.75 focusing tube diameter was selected for the first and the second block, respectively. Each focusing tube was purchased from ROCTEC$^\circledR$ (Kennametal, Latrobe, PA, USA). The experimental design was generated according to the statistical software MINITAB$^\circledR$. The resulting design matrix in the randomized run order is reported in Table 5.

**Table 5.** $2_{IV}^{7-3}$ design matrix.

| Run Order | $d_n$ | $p$ | $sod$ | $\dot{m}_a$ | $v$ | $t$ | $\vartheta$ |
|---|---|---|---|---|---|---|---|
| 1 | 0.33 | 260 | 50 | 200 | 1.11 | 15 | 90 |
| 2 | 0.33 | 190 | 100 | 50 | 2.22 | 25 | 90 |
| 3 | 0.33 | 260 | 100 | 200 | 1.11 | 25 | 60 |
| 4 | 0.33 | 190 | 100 | 200 | 2.22 | 15 | 90 |
| 5 | 0.33 | 190 | 50 | 200 | 2.22 | 25 | 60 |
| 6 | 0.33 | 260 | 50 | 50 | 2.22 | 25 | 90 |
| 7 | 0.33 | 260 | 100 | 50 | 2.22 | 15 | 60 |
| 8 | 0.33 | 190 | 50 | 50 | 1.11 | 15 | 60 |
| 9 | 0.25 | 260 | 100 | 200 | 2.22 | 25 | 90 |
| 10 | 0.25 | 190 | 100 | 200 | 1.11 | 15 | 60 |
| 11 | 0.25 | 260 | 100 | 50 | 1.11 | 15 | 90 |
| 12 | 0.25 | 190 | 100 | 50 | 2.22 | 25 | 60 |
| 13 | 0.25 | 260 | 50 | 50 | 1.11 | 25 | 60 |
| 14 | 0.25 | 260 | 50 | 200 | 2.22 | 15 | 60 |
| 15 | 0.25 | 190 | 50 | 50 | 2.22 | 15 | 90 |
| 16 | 0.25 | 190 | 50 | 200 | 1.11 | 25 | 90 |

*2.4. Data Analysis*

Since the cutting efficiency $\varepsilon$ was defined as a fraction Equation (7), it assumes values in the interval {0,1} by definition. Linear regression applied to fractions modelling results in three different problems [34]: non-constant error variance, non-normally distributed residuals, and out-of-boundary predictions. In fact, linear models can predict values for response variable higher than 1 or lower than 0, which are meaningless in this kind of problems. For this, cutting efficiency data were modelled with logistic regression. This model does not consider any specific probability distribution of the error term, and the non-constant variance of the error term can be easily handled [34]. Indeed, it naturally constrains the response variable into the interval {0,1}. On the other hand, the meaning and physical interpretation of the model's coefficients are less intuitive than for the case of linear regression models. The data analysis was performed according to the following steps:

1. Exploratory graphical analysis: data were represented according to the main effect plots, as well as two factors interaction plots, to visually investigate the most influential effects;
2. Multivariable logistic model was built: the odds ratio of each predictor to the response variable (i.e., the cutting efficiency) was estimated;
3. An Analysis of deviance was performed to identify significant factors;
4. For each odds ratio, the 95% CI was estimated.

## 3. Results

*3.1. Preliminary Experiment*

Results of the preliminary cutting test is shown in Figure 5, where cutting efficiency is plotted at different levels of water pressure. A full separation cut was not achieved when water pressure was below 150 MPa. For pressure higher than 175 MPa, a separation cut was achieved. This result might be connected to the fact that waterjet needs enough power to penetrate through wheat straws.

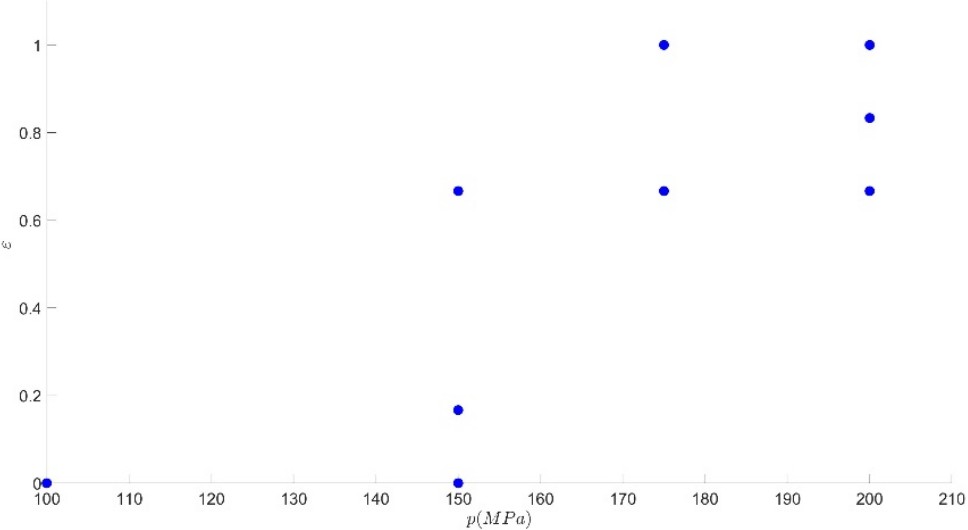

**Figure 5.** Scatterplot of the cutting efficiency at different levels of water pressure.

### 3.2. Stasitically Designed Experiment

As a starting point, a graphical evaluation of the response variable was performed: Figure 6 shows the main effects plots. Few factors seem to influence the average response level, i.e., primary orifice diameter, straw layer thickness, and water pressure seem to be the most influential among the considered factors.

The most potentially significant two-factor interaction plot was identified [33], and its graphical representation is shown in Figure 7. Since the interaction between standoff distance and traverse velocity may be significant, it has been considered in subsequent statistical analysis.

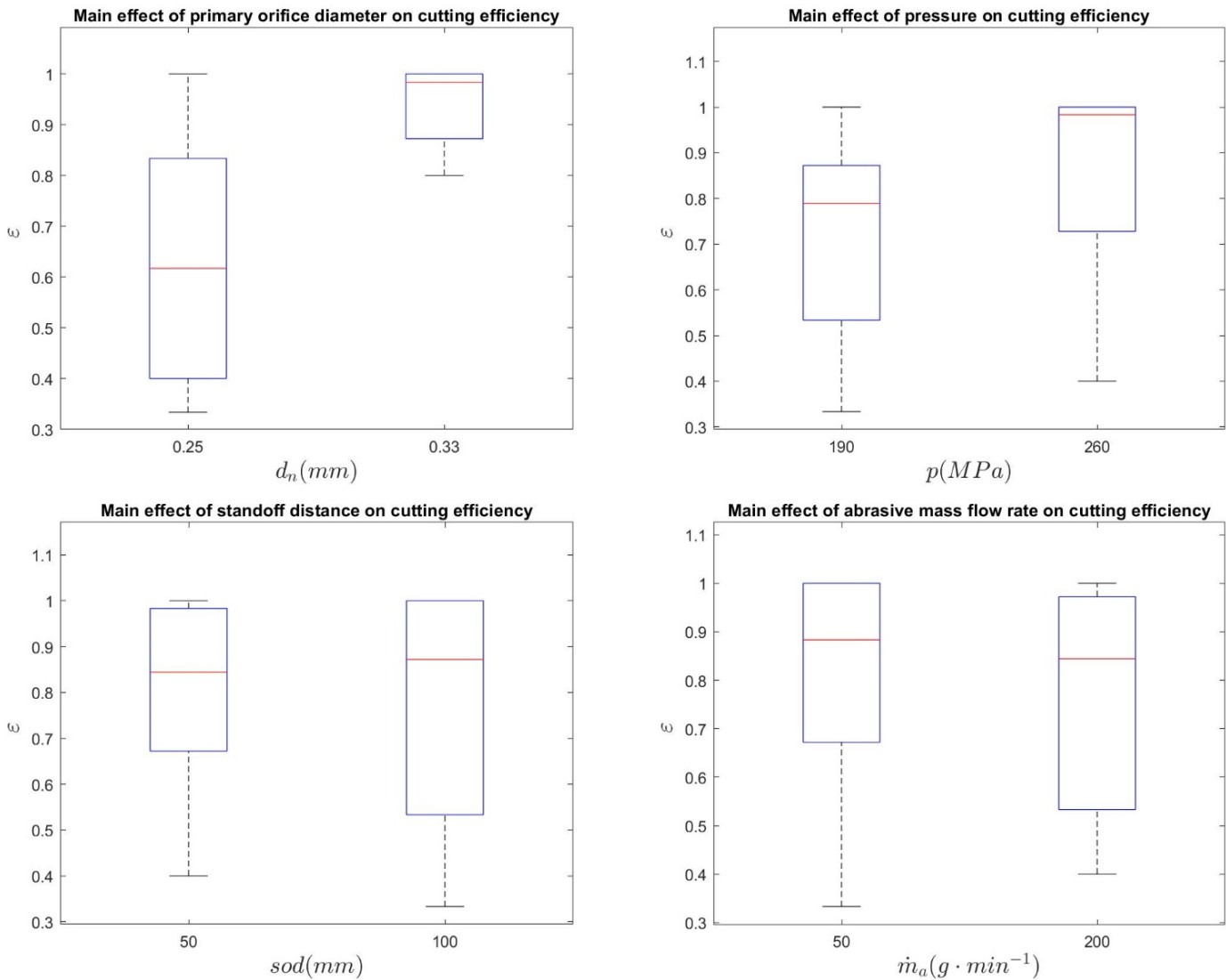

**Figure 6.** *Cont.*

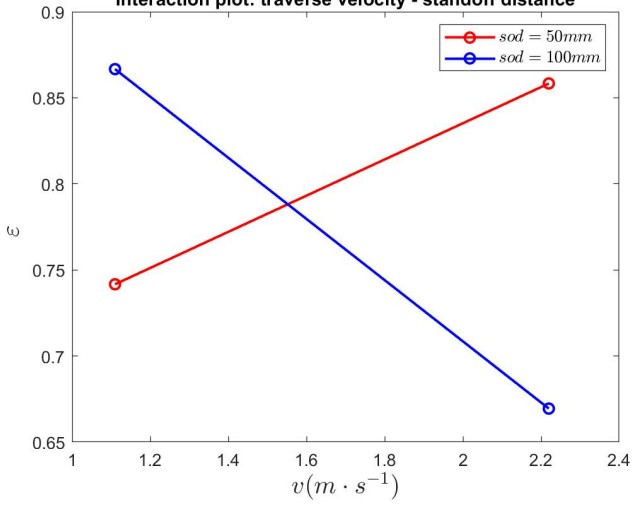

**Figure 6.** Main effects plots.

**Figure 7.** Two factor interaction plot between standoff distance and traverse velocity.

### 3.3. Data Analysis

The fitting of a logistic model with experimental data was evaluated for the following selected parameters: primary orifice diameter, water pressure, standoff distance, abrasive mass flow rate, traverse velocity, thickness, orientation, and the two-factor interaction between standoff distance and traverse velocity. Analysis of deviance [34] was performed to look for significant factors. The overall results are reported in Table 6 (terms were added sequentially), and three factors were found to exhibit more evident significance in the model ($p$-value < 0.05).

**Table 6.** Analysis of deviance table.

| Source | Df | Deviance | Residual Df | Residual Deviance | $p$-Value |
|---|---|---|---|---|---|
| Null | | | 15 | 138.573 | |
| Primary orifice diameter | 1 | 67.931 | 14 | 70.462 | $<2.2 \times 10^{-16}$ |
| Water pressure | 1 | 12.352 | 13 | 58.290 | 0.0004404 |
| Standoff distance | 1 | 0.856 | 12 | 57.434 | 0.3547515 |
| Traverse velocity | 1 | 1.153 | 11 | 56.281 | 0.2830232 |
| Abrasive mass flow rate | 1 | 0.863 | 10 | 55.418 | 0.3528519 |
| Thickness | 1 | 42.526 | 9 | 12.892 | $6.976 \times 10^{-11}$ |
| Orientation | 1 | 0.009 | 8 | 12.883 | 0.9238731 |
| Standoff distance $\times$ traverse velocity | 1 | 0.030 | 7 | 12.853 | 0.8633043 |

Finally, a multivariable logistic model was built including significant factors: primary orifice diameter, water pressure, and sample thickness. Estimated effects and their 95% CI are shown in Table 7.

**Table 7.** Estimated effects and 95% CI.

| Source | Estimate | 95% Confidence Interval | | $p$-value |
|---|---|---|---|---|
| Intercept | 1.1764 | 0.5698577 | 1.850685 | 0.000283 |
| Primary orifice diameter | 2.5616 | 1.9230061 | 3.267258 | $6.04 \times 10^{-14}$ |
| Water pressure | 1.0869 | 0.5179666 | 1.680748 | 0.000238 |
| Thickness | −2.0493 | −2.7868986 | −1.381458 | $9.12 \times 10^{-9}$ |

From the data analysis results, the following statements can be drawn:

- Primary orifice diameter and water pressure, in Table 7, are both significant factors: moving from low level to high level the cutting efficiency increases. This result agrees with [27,28];
- Straw orientation and abrasive mass flow rate were found to have poor significance in the model, likely indicating that the cutting efficiency may not be based on material erosion due to the interaction between abrasive particles and straw tissue but rather on the impulsive impact of the jet on straw;
- Standoff distance was found to have poor significance in the model: in all treatment combinations (with a ratio $sod/d_n < 400$), the waterjet kinetic energy was sufficiently high to cut through the target straw.

## 4. Discussion

In the waterjet technology field, the hydraulic power of the jet, $P_{hydr}$, is directly connected to the hydraulic parameters Equation (6). Power is a fundamental quantity that explains jet cutting capability of many soft materials and fibre-composite soft materials [11] that are subject to the shearing and pulling out of fibres, depending on the level of the jet power [35,36]. Similarly, wheat straws can be classified as a fibre-composite soft material. In fact, plant stem structure consists of cavities filled with liquid and air encapsulated in a solid matrix, with considerably long transverse fibers made of chains of cellulose and lignin [37]. When dried, the structure consists of an assembly of long chain fibres immersed in a soft material matrix [38].

Figure 8 shows a comparison between an untreated straw cross section (Figure 8a) and the cut surface of a wheat straw after AWJ cutting (Figure 8c). The inner region of the straws is characterized by a neat separation, while the outer region shows the presence of natural fibres (Figure 8d), which appear to be largely peeled off from their original position (Figure 8b). Possibly, as soon as the waterjet approaches the outer surface of the straw, a shear stress take place that may induce the fibre deformation until it fails. However, as jet penetrates the sample, it loses its power, indeed, since the standoff distance increases, jet coherence decreases resulting in higher spreading of jet energy. Consequently, according to the initial hydraulic power and straws thickness, a gradually diminishing jet capability has been observed throughout the sample thickness. Partially cut straws were examined in the bottom region each sample, as shown in Figure 9. In the same figure, fibres delamination appears to be clearly evident. This mechanism was found to be a characteristics phenomenon in AWJ cutting of composites material [10] such as bio-fibre reinforced composites. The authors of [39,40] showed that the basic mechanism of fibres delamination increased with increase in standoff distance, due to the expansion of the jet.

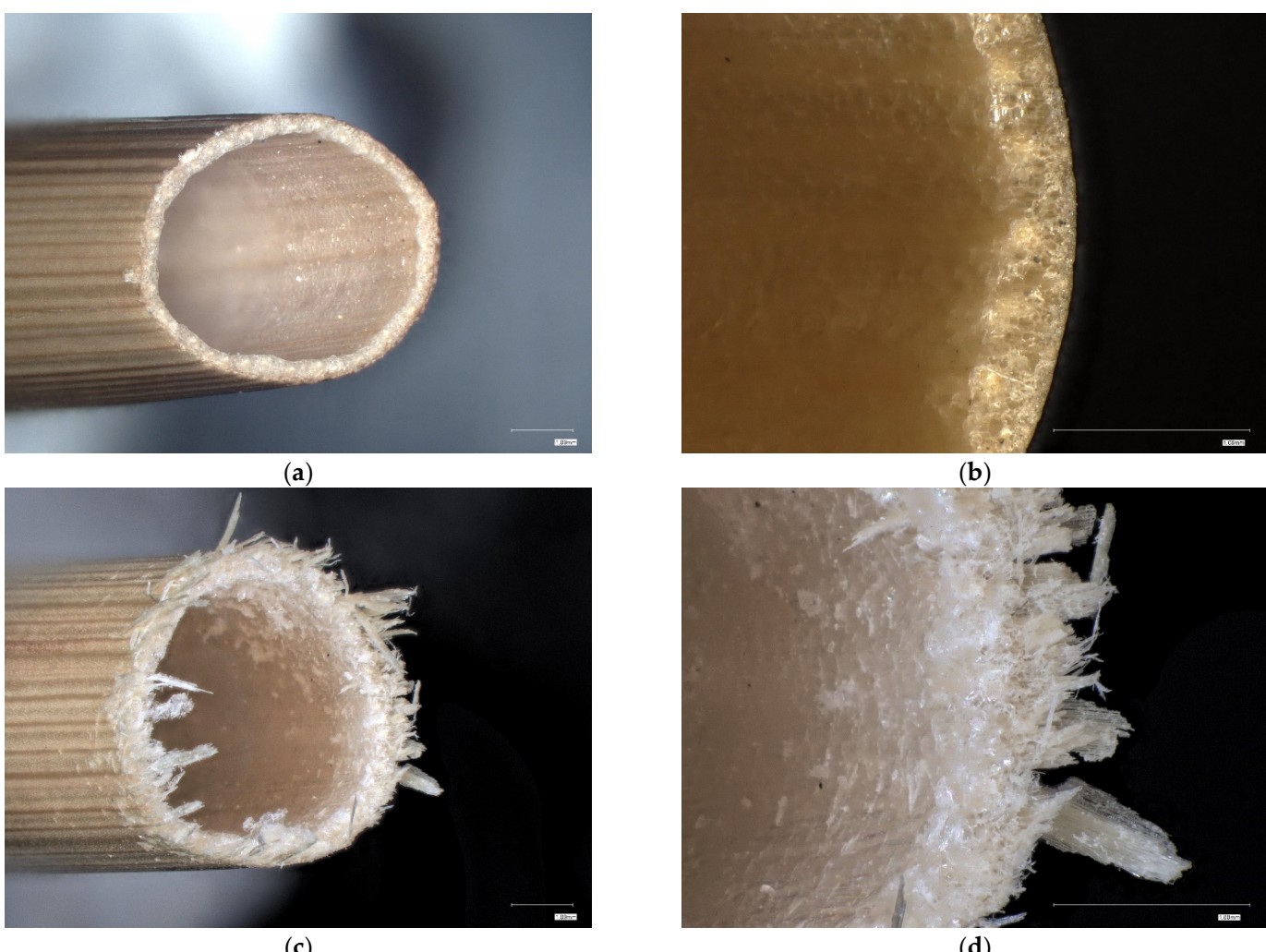

**Figure 8.** Profile cross section of (**a**) untreated straw (marker shows 1 mm). (**b**) Particular of the untreated straw cross section structure. (**c**) Straw after abrasive waterjet cutting, when $d_n$ = 0.33 mm, $p$ = 260 MPa, *sod* = 50 mm, $v$ = 1.11 m·s$^{-1}$, and $\dot{m}_a$ = 200 g·min$^{-1}$. (**d**) Particular of the straw cross section structure after AWJ cutting.

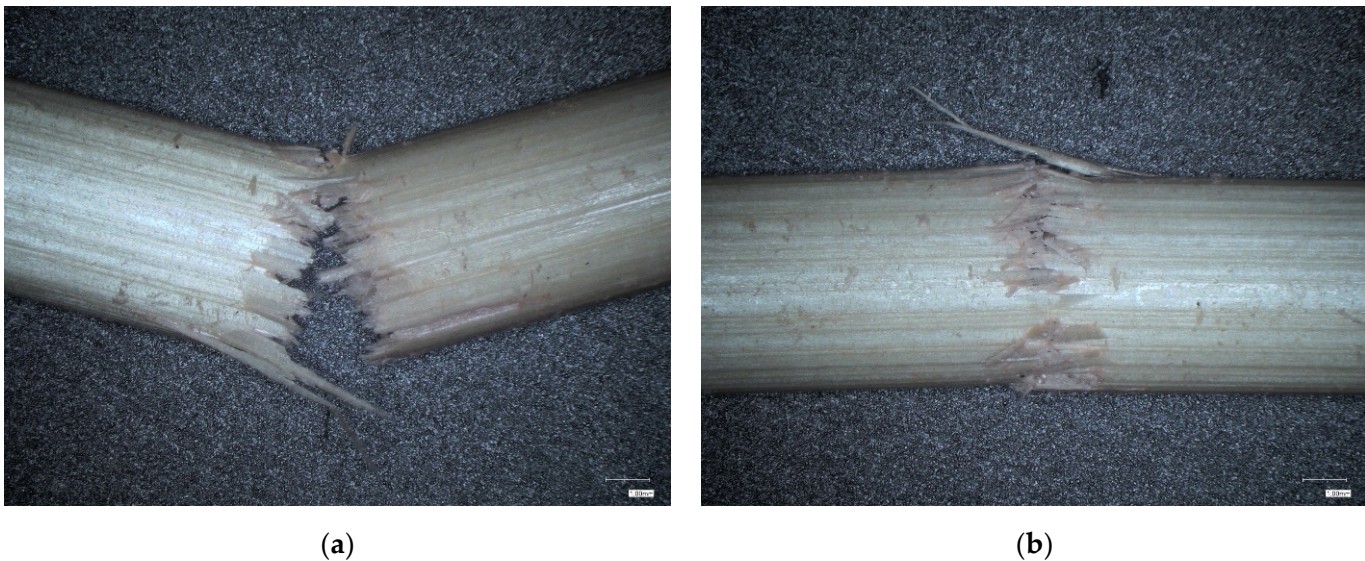

(**a**)　　　　　　　　　　　　　　　　(**b**)

**Figure 9.** Partially cut straw. (**a**) Detail of the cut region. (**b**) Detail of the uncut region.

Since water pressure and primary orifice diameter were found to be significant, with the hydraulic power being a combination of these two parameters, it turns out that jet hydraulic power can be an adequate quantity to explain the cutting capability of wheat straws. For this purpose, a logistic regression model was fitted to experimental data, considering both jet power and sample thickness as predictors and the cutting efficiency as the response variable. Results are shown in Table 8.

**Table 8.** Results from logistic regression model to the cutting efficiency $\varepsilon$.

| Source | Symbol | Coefficient | Std. Error | Z Score | *p*-Value |
|--------|--------|-------------|------------|---------|-----------|
| Intercept | | 1.6781808 | 0.8644443 | 1.941 | 0.0522 |
| Power | $P_{\text{hydr}}$ | 0.0007238 | 0.0001027 | 7.049 | $1.80 \times 10^{-12}$ |
| Thickness | $t$ | −0.2024999 | 0.0355011 | −5.704 | $1.17 \times 10^{-8}$ |

A regression equation was obtained in (8):

$$\log_e \left( \frac{\varepsilon}{1 - \varepsilon} \right) = 1.68 + 7.2 \times 10^{-4} P_{\text{hydr}} - 0.20t \tag{8}$$

where, $P_{\text{hydr}}$ is the jet hydraulic power (W), while $t$ is the sample thickness (mm). Measured cutting efficiencies are shown in Figure 10 along with predicted values, according to the logistic model Equation (8).

Model predictive capability was evaluated by performing three additional runs, as shown in Table 9. Predicted cutting efficiency ($\varepsilon$) was found to be in agreement (95% confidence interval) with measured values.

Results show a progress in understanding waterjet cutting capability of crop residues, with respect to the existing scientific and technical literature. In fact, in [28] a very high cutting capability of wheat residues has been achieved in two passes, even though the hydraulic power of the pure waterjet was extremely high (13 kW) since water pressure, $p$ = 380 MPa and orifice diameter, $d_{\text{n}}$ = 0.30 mm. We proved that the use of a minimum quantity of abrasive increases jet-cutting capability, in the face of a lower hydraulic power requirement (6 kW), compared to the pure waterjet cutting.

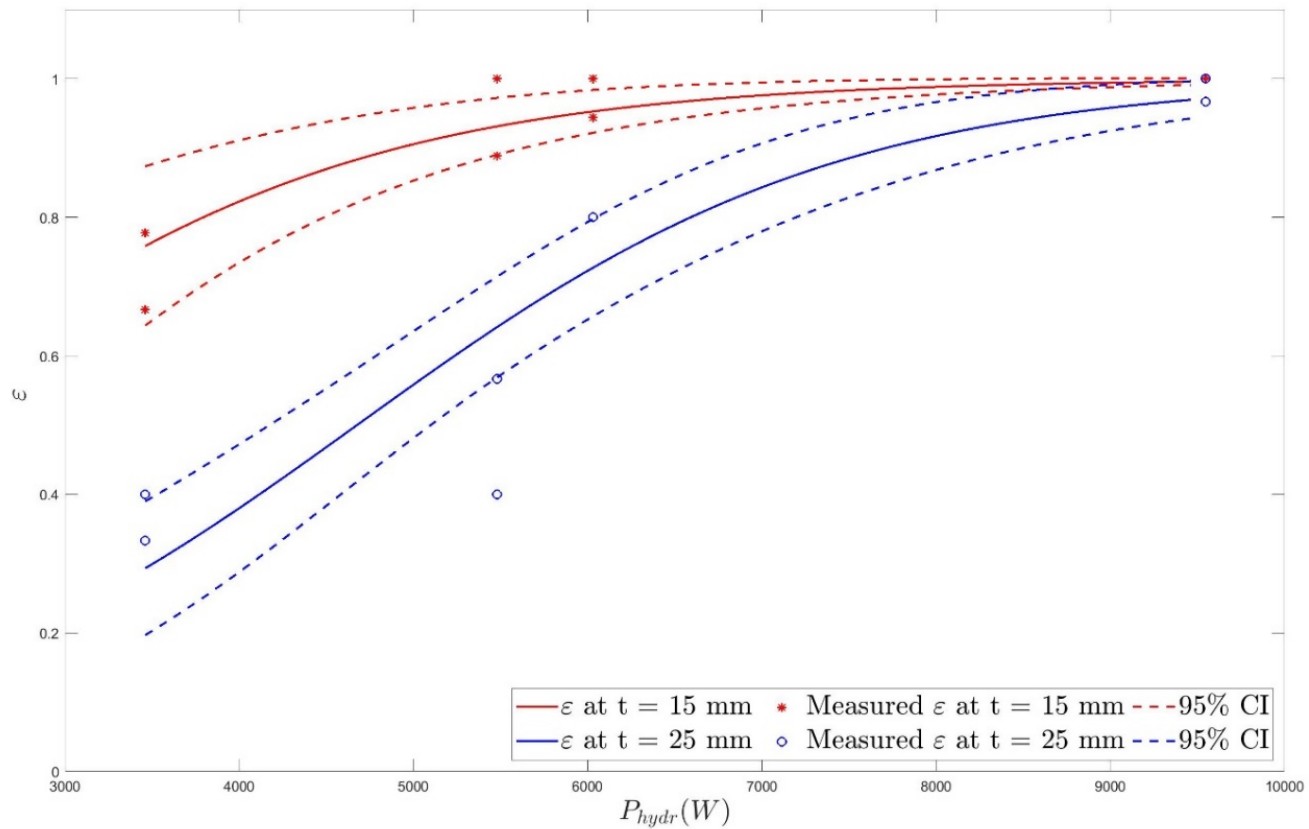

**Figure 10.** Plot of measured and predicted cutting efficiency against jet hydraulic power $P_{\text{hydr}}$.

**Table 9.** Model validation.

| Run | $d_{\text{n}}$ (mm) | $p$ (MPa) | $P_{\text{hydr}}$ (W) | $t$ (mm) | Measured | Predicted | 95% Confidence Interval | |
|---|---|---|---|---|---|---|---|---|
| 1 | 0.25 | 200 | 3730 | 15 | 0.61 | 0.79 | 0.69 | 0.89 |
| 2 | 0.25 | 260 | 5250 | 25 | 0.47 | 0.43 | 0.34 | 0.52 |
| 3 | 0.30 | 225 | 6390 | 20 | 0.88 | 0.90 | 0.86 | 0.94 |

Considering a primary orifice with diameter $d_{\text{n}}$ = 0.30 mm and water pressure, $p$ = 225 MPa, the emerging jet would have a sufficiently high hydraulic power (6390 W) to achieve 90% of cutting efficiency (Equation (8)), under the presence of 20 mm residue thickness, together with a quantity of abrasive mass flow rate, $\dot{m}_{\text{a}} = 50$ g·min$^{-1}$. However, power requirement is clearly higher than the power consumption of a commercial no-till seeding unit. In the latter case, the average power consumption for the seeding process can reach values up to 6 kW [4] for the whole seeding process.

Results show that when residue thickness is about 25 mm, the jet cutting capability is reduced. This is because a large fraction of the jet energy is used in the cutting process; a small quantity of energy remains available for soil opening. For this reason, we presume that higher hydraulic power would be required to carry out the seed bed opening. In fact, in [26] results showed that 5 kW of hydraulic power was required to penetrate up to 70–90 mm.

Further studies will be focused on the impact that abrasive material might have on soil as well as on the seedbed environment. Since the use of natural abrasive like river sand may be a valuable alternative to garnet abrasive, as suggested in [31], further investigations will be executed. Furthermore, the effect of jet on agricultural soil properties will be investigated.

## 5. Conclusions

In this work, the abrasive waterjet cutting of wheat straws was investigated, from the perspective of exploiting the technology for no-till cutting tools. The main results of this study are presented as follows:

- The most significant process parameters influencing the straws cutting efficiency were identified: primary orifice diameter, as well as water pressure, were found to be the most influential process parameters, along with straws coverage thickness.
- Standoff and traverse velocity were found to be not significant. Since the jet was able to cut at standoff distance up to 150 mm and velocity of 2.2 m·s$^{-1}$, a field application seems to be reasonable.
- Jet hydraulic power might be the most valuable quantity to explain the cutting efficiency.
- Power and water consumptions were assessed considering a reference scenario of maize direct seeding. The required hydraulic power for cutting operations was estimated to range from 5 kW up to 7.5 kW in presence of a residue coverage thickness that ranges from 15 mm to 25 mm. Indeed, an average water consumption of about 30 L·ha$^{-1}$ per nozzle was estimated.

The study shows that a small quantity of abrasive powder (0.5 g·m$^{-1}$) allows one to increase jet cutting capability of wheat straws, as well as to decrease the required hydraulic power, compared to the case of pure waterjet cutting. Considering as an example a row seeding process (maize), where the inter row seeding spacing is 0.75 m and there is a working velocity of 2.0 m·s$^{-1}$, a field area of 1 ha, a row length of 100 m, and two nozzles per meter of working width, the total amount of abrasive consumption would be o about 0.56 g·m$^{-2}$ (5.6 kg·ha$^{-1}$).

Results have confirmed that waterjet cutting might be a valuable alternative method for residue management, in a no-till scenario, resulting in high cutting performance in presence of a thick mulch of crop residues, even though results must be verified in a field experiment. Waterjet cutting might increase the process performance in terms of high cutting efficiency and can reduce the tractor's engine power necessary to carry out the operation, compared to standard mechanical residue management methods. However, high pressure is required (at least 200 MPa) to obtain valuable cutting performance. This might impact the technological development, due to high cost and consumptions, related to the water pump. Future experimental studies are needed to investigate and characterize the waterjet and soil interaction, in terms of furrow opening capability and geometry, as well as the residual effects on soil physical properties, and to evaluate the savings in terms of engine power, traction force, and fuel required by the tractor involved in the operation.

**Author Contributions:** Conceptualization, F.P., A.C., and R.O.; methodology, F.P. and V.M.; formal analysis, F.P.; investigation, F.P.; data curation, F.P.; writing—original draft preparation, F.P.; writing—review and editing, A.C., R.O., and M.A.; supervision, M.A. and M.M.; funding acquisition, M.M. All authors have read and agreed to the published version of the manuscript.

**Funding:** This research received no external funding.

**Institutional Review Board Statement:** Not applicable.

**Data Availability Statement:** Not applicable.

**Acknowledgments:** The authors gratefully acknowledge the support of Regione Emilia-Romagna administration.

**Conflicts of Interest:** The authors declare no conflict of interest.

## Nomenclature of the Abrasive Waterjet Cutting Technology

| | | |
|---|---|---|
| $p$ | MPa | Water pressure |
| $d_n$ | mm | Primary orifice diameter |
| $\rho$ | kg·m$^{-3}$ | Water density |
| $v_{th}$ | m·s$^{-1}$ | Waterjet theoretical velocity |
| $n, L$ | -, MPa | Constants |
| $v_j$ | m·s$^{-1}$ | Real jet velocity |
| $\psi$ | - | Compressibility coefficient |
| $c_v$ | - | Velocity coefficient |
| $c_d$ | - | Discharge coefficient |
| $\dot{m}_w$ | kg·s$^{-1}$ | Water mass flow rate |
| $Q_w$ | m$^3$·s$^{-1}$ | Water volume flow rate |
| $S_n$ | m$^2$ | Jet cross-sectional area of the orifice |
| $P_{hydr}$ | W | Jet hydraulic power |
| $\dot{m}_a$ | kg·s$^{-1}$ | Abrasive mass flow rate |
| $d_f$ | mm | Focuser tube diameter |
| sod | mm | Standoff distance |
| $v$ | m·s$^{-1}$ | Traverse velocity |
| $\vartheta$ | ° | Sample orientation |
| $t$ | mm | Sample thickness |
| $\varepsilon$ | - | Cutting efficiency |

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
