# Peer review of "Experimental Study of Abrasive Waterjet Cutting for Managing Residues in No-Tillage Techniques"

_agriculture, doi:10.3390/agriculture11050392_

Round 1
Reviewer 1 Report
The article presents an extended study, including experimental research, of abrasive water jet cutting with application in conservation agriculture.
Research campaign appears highly detailed.
Concerning Ishikawa diagram (fig. 2) the 6M approach does not appear, it is recommended to revise it.
Concerning the relevance of practical application of this proposed cutting method, a more clear rationale should be detailed. Why should one apply this method instead of the classical one.
Reviewer 2 Report
Please see comments on attached manuscript.
Comments:
This is an important experimental development but has some serious issues that will keep it moving forward in the literature. The major item is the inconsistency in several areas of the manuscript such as nomenclature (then following them in the manuscript), units (the use of “/”), citation (several errors) and references (not listing publishers or incomplete).
There is also the issue of using the limited experimental data and showing regression curves outside the data limits (Figure 10-12). It becomes speculation at best and does not really provide much insights. Also, consistent axis labels and units are needed throughout the manuscript. The calculations in Section 4.1 seem to be a stretch for the manuscript and if needed, I would suggest this section be moved to an appendix.
Issues that need to be discussed and addressed:
In lines 198-199 the added abrasive material is specified but no discussion is given on the impact this material might have on the soil and close proximity of the seed. Since this is am important ingredient to the success of this system, some discussion is warrantied.
One of the reasons that residue is effective is it slows the rain impact and limits crushing that could slow water penetration (compaction of the surface soil layer). There needs to be acknowledgement that high pressure water could have an adverse effect on the soil and soil-to-seed contact. While this seems like it is outside the scope of the manuscript the field application will need to consider and address.
Final unless the straw is secured in the field how will the jet not just blow the residue around and not cut? Give some insight in the field applicator.

Reviewer 3 Report
The paper topic is interesting but needs the above suggestions:
Abstract: The abstract is expected to include a brief digest of the research, that is, new methods, results, concepts, and conclusions only. The abstract needs to be more focused and achievements needs mentioned clearly. At the moment abstract is more like an introduction than abstract. Please add some information from the conclusion (quantifications).
Introduction is weak point of this paper and based on old references. I personally feel that this part of paper is not concise enough from a reader’s perspective. Introduction must provide a comprehensive critical review of recent developments in a specific area or theme that is within the scope of the journal (advanced manufacturing), not only a list of published studies or a bibliometric one. Introduction is expected to have an extensive literature review followed by an in-depth and critical analysis of the state of the art. References section should be extensive about information connecting with AWJ, PWJ and applications. I suggest add information to better describe what other researchers have done in this area.
The strengths and limitations of the applied approach should be clearly identified for the readers of the paper.
Descriptions with defects on micrographs should be added.
The discussion is shallow and needs more details, the observations and future trends. This chapter should be connected with others published papers.
Some of the bullet points on the conclusion are simplistic; Please try to emphasize your novelty, put some quantifications, and comment on the limitations. This is a very common way to write conclusions for a learned academic journal. The conclusions should highlight the novelty and advance in understanding presented in the work.
Round 2
Reviewer 2 Report
Some of the axis of graph are very difficult to read - please increase the font size
Reviewer 3 Report
The Authors improved the paper but only in points where the Authors wanted, in the present form paper is below journal standard. The answer that is lack of AWJ literature in agriculture is not serious, discussion is weak.
